# Heterogeneity and Differentiation Trajectories of Infiltrating CD8+ T Cells in Lung Adenocarcinoma

**DOI:** 10.3390/cancers14215183

**Published:** 2022-10-22

**Authors:** Xiaojie Song, Guanghui Zhao, Guangqiang Wang, Haidong Gao

**Affiliations:** 1Department of Respiratory and Critical Care Medicine, Qilu Hospital (Qingdao), Cheeloo College of Medicine, Shandong University, Qingdao 266035, China; 2Medical Laboratory Center, Qilu Hospital (Qingdao), Cheeloo College of Medicine, Shandong University, Qingdao 266035, China; 3Oncology Laboratory, Qilu Hospital (Qingdao), Cheeloo College of Medicine, Shandong University, Qingdao 266035, China; 4Department of Breast Surgery, Qilu Hospital (Qingdao), Cheeloo College of Medicine, Shandong University, Qingdao 266035, China

**Keywords:** lung adenocarcinoma, CD8+ T cells, heterogeneity, differentiation trajectory, functional exhaustion

## Abstract

**Simple Summary:**

CD8+ T cells infiltrating the tumor microenvironment (TME) of lung adenocarcinoma (LUAD) play a crucial role in establishing anti-tumor immunotherapy. The number of CD8+ T cells affects the treatment response, but their functional status plays a more critical role, and this global landscape is still unclear. We divided CD8+ T cells into ten subsets by analyzing a LUAD single-cell dataset. The dynamic process of cell differentiation and functional exhaustion of CD8+ T cells was further discussed, and potential biomarkers in this process were screened. This study deepens the understanding of the heterogeneity of infiltrating CD8+ T cells in LUAD, and the prognostic marker provides a new target for targeted therapy and immunotherapy in LUAD patients.

**Abstract:**

CD8+ T cells infiltrating the tumor microenvironment (TME) of lung adenocarcinoma (LUAD) are critical for establishing antitumor immunity. Nevertheless, the global landscape of their numbers, functional status, and differentiation trajectories remains unclear. In the single-cell RNA-sequencing (scRNA-seq) dataset GSE131907 of LUAD, the CD8+T cells were selected for TSNE clustering, and the results showed that they could be divided into ten subsets. The cell differentiation trajectory showed the presence of abundant transition-state CD8+ T cells during the differentiation of naive-like CD8+ T cells into cytotoxic CD8+ T cells and exhausted CD8+ T cells. The differentially expressed marker genes among subsets were used to construct the gene signature matrix, and the proportion of each subset was identified and calculated in The Cancer Genome Atlas (TCGA) samples. Survival analysis showed that the higher the proportion of the exhausted CD8+ T lymphocyte (ETL) subset, the shorter the overall survival (OS) time of LUAD patients (*p* = 0.0098). A total of 61 genes were obtained by intersecting the differentially expressed genes (DEGs) of the ETL subset, and the DEGs of the TCGA samples were divided into a high and a low group according to the proportion of the ETL subset. Through protein interaction network analysis and survival analysis, four hub genes that can significantly affect the prognosis of LUAD patients were finally screened, and RT-qPCR and Western blot verified the differential expression of the above four genes. Our study further deepens the understanding of the heterogeneity and functional exhaustion of infiltrating CD8+ T cells in LUAD. The screened prognostic marker genes provide potential targets for targeted therapy and immunotherapy in LUAD patients.

## 1. Introduction

According to the latest cancer burden data released by the International Agency for Research on Cancer (IARC), there were 2.2 million new lung cancer cases worldwide in 2020, making it the second-largest new cancer in the world. However, it is astonishing that among the 9.96 million cancer deaths in that year, lung cancer accounted for 1.8 million cases, far exceeding other cancer types [1]. This vast difference in the number of morbidities and deaths indicates that the clinical treatment of lung cancer seems to have entered a bottleneck period. The rise of tumor immunotherapy has made significant progress in treating lung adenocarcinoma. About 20% of lung cancer patients achieve significant clinical remission [2], but most lung cancer patients still do not respond to immunotherapy or do not respond significantly [3].

The number of infiltrating CD8+ T cells in the TME partly explains this variability [4], but their functional status plays a more critical role [5]. It is now clear that infiltrating CD8+ T cells in tumor tissue exist in a naive-like, effector, resident memory, or exhausted state [6]. The distribution and evolution of these different states of CD8+ T cells in LUAD are unclear, especially the transition process between these states.

Through single-cell sequencing of samples from 14 newly treated patients with non-small cell lung cancer (NSCLC), Guo et al. observed two cell clusters in the pre-exhausted state during the depletion of tumor-infiltrating CD8+ T cells. A high ratio of pre-exhausted T cells to exhausted T cells is associated with a better prognosis in LUAD [7]. These results suggest that the functional status of CD8+ T cells affects LUAD patients’ prognosis. Kim et al. performed single-cell sequencing of 208,506 cells from 58 samples of 44 patients with LUAD, covering normal tissue, early-stage LUAD, and advanced metastatic LUAD, delineating the unique single-cell transcriptome profile of metastatic lung adenocarcinoma [8]. Among them, the authors’ analysis confirmed the differential distribution of CD8+T cells in naive, cytotoxic, or exhausted states in LUAD tissues and finally suggested that the direction of tumor immunity should be towards immune suppression in LUAD. However, more studies have shown that CD8+ T cells do not exist in a simple ternary state (naive, cytotoxic, or exhausted) within the tumor, but rather as a continuous transitional process [6,7], and the original analysis of CD8+ T cells is inadequate and deserves further development.

The authors have shared the single-cell sequencing data generated from this study to the GEO database (GSE131907). We started with this data set because of the large sample size in this study and the complete staging of the patients with LUAD. Combined with TCGA samples and our clinical samples, we focused on the global landscape of infiltrating CD8+ T cells in the LUAD, including number, subgroups, distribution ratio, functional status, and dynamic evolution. This will deepen our understanding of the heterogeneity and functional exhaustion of infiltrating CD8+ T cells in LUAD. Finally, four DEGs that significantly affect the prognosis of LUAD patients among different subgroups were screened and verified, providing potential targets for blocking CD8+ T cell function exhaustion and improving the response rate of immunotherapy. To our knowledge, this is the first description to focus on the dynamic evolution of LUAD CD8+ T cells at the single-cell level.

## 2. Materials and Methods

### 2.1. LUAD scRNA-Seq Data Download and Processing

The LUAD scRNA-seq dataset GSE131907 (IlluminaHiSeq2500) was downloaded from the GEO database (https://www.ncbi.nlm.nih.gov/geo/, accessed on 1 June 2022), which contains 58 specimens from 44 LUAD patients of 208,506 cells. Cell data of tumor tissues were selected, and 57,222 cells were obtained. The single-cell data were filtered and dimensionally reduced using the R ‘Seurat’ and the ‘dplyr’ packages. The filtering criteria were to delete cells with fewer than 50 genes measured and cells with mitochondrial gene percentages greater than or equal to 5%. Genes expressed in 3 or fewer samples were also filtered out, resulting in 41,910 cells.

### 2.2. Extraction of CD8+ T cells

Merged data were normalized through log-normalization, and the top 2000 genes with significant coefficients of variation were selected according to the variance value for PCA analysis. According to the *p* value, the top 20 PC components were chosen for TSNE clustering. Clusters rich in CD8+ T cells were selected according to marker genes CD8A and CD8B and then filtered in order to select those with CD8A and CD8B expression levels above 0, and the samples with CD4 expression levels greater than 0 were deleted. A total of 5753 cells were obtained.

### 2.3. Identification of CD8+T Cell Subsets and Analysis of Marker Gene Expression

TSNE clustering was performed on the finally obtained CD8+ T cells, and ten CD8+ T cell subsets were obtained. The expression levels of known CD8+ T cell functional status markers were analyzed in ten subsets.

Naive-like CD8+ T cells markers include chemokine receptor 7 (CCR7), lymphoid enhancer-binding factor 1 (LEF1), transcription factor 7 (TCF7), and L-selectin (SELL).

Cytotoxic CD8+ T cell markers include perforin 1 (PRF1), granzyme A (GZMA), granzyme B (GZMB), granzyme H (GZMH), natural killer cell granule protein 7 (NKG7), interferon gamma (IFNG), granulysin (GNLY), killer cell lectin-like receptor K1 (KLRK1), killer cell lectin-like receptor B1 (KLRB1), killer cell lectin-like receptor subfamily D member 1 (KLRD1), cathepsin W (CTSW), and cystatin F (CST7).

Exhausted CD8+ T cells markers include cytotoxic T lymphocyte-associated protein 4 (CTLA4), programmed cell death 1 (PDCD1), lymphocyte activation gene 3 (LAG3), hepatitis A virus cellular receptor 2 (HAVCR2), and T cell immunoreceptor with Ig and ITIM domains (TIGIT).

The markers of short-lived effector cells (SLECs) and memory precursor effector cells (MPECs) include T-Box Transcription Factor 21 (TBX21 or T-Bet), Killer Cell Lectin-like Receptor G1 (KLRG1), C-X3-C Motif Chemokine Receptor 1 (CX3CR1) and CD127 (IL7R). The markers of progenitor or terminally exhausted CD8+ T cells include Transcription Factor 1 (TCF1 or TCF7), Thymocyte Selection Associated High Mobility Group Box (TOX) and PD1.

### 2.4. Distribution and Survival Analysis of CD8+ T Cell Subsets in TCGA Samples

Based on the CIBERSORT algorithm, a signature matrix was constructed using the differentially expressed genes (DEGs) of 10 CD8+ T cell subsets in the single-cell sequencing data. It was used as the reference matrix of immune infiltration. The screening criteria for DEGs were |log2 fold changes (FC)| > 0.5, and the adjusted *p* value was less than 0.05. The proportions of different CD8+ T cell subsets in TCGA samples were calculated, and their impact on the prognosis of LUAD patients was analyzed. Briefly, the mRNA expression profiles of LUAD samples were downloaded from the UCSC Xena (https://xenabrowser.net/, accessed on 4 June 2022) database. First, the rank of the differential marker gene expression values was normalized. Then, the cumulative distribution function was used to calculate the enrichment scores, representing the enrichment degree of the gene set in the given samples, which can be regarded as the proportion of immune cells. The TCGA samples were divided into high and low ratio groups by the median, and survival analysis was performed on the two groups of patients using the R ‘survival’ package.

### 2.5. Correlation Analysis between Exhausted CD8+T Cells and Clinicopathological Characteristics

The correlation between the number of exhausted CD8+ T cells and the TNM stage and AJCC stage of patients was analyzed. The distribution of exhausted CD8+ T cells in different stages was observed.

### 2.6. Differentiation Trajectories of CD8+ T Cell Subsets and Distribution of Marker Genes

Cell differentiation trajectory analysis was performed using the R ‘monocle’ package, which arranges cells in simulated chronological order. The distribution of some marker genes is displayed by pseudo time.

### 2.7. Network Analysis of Different CD8+ T Cell Subsets

Ligand–receptor relationships were taken from the literature [9]. The cell-to-cell interaction network analysis was performed through the Python ‘CellPhoneDB’ package, and the relationships with significant differences (*p* < 0.05) in different CD8+ T cell subsets were obtained. The relationship data of transcription factors (TFs) and target genes were downloaded from the TRRUST database (https://www.grnpedia.org/trrust/, accessed on 4 July 2022). The TFs and genes in the ligand–receptor network in different subsets were intersected to obtain the differentially expressed TFs in CD8+ T cell subsets.

### 2.8. Analysis of DEGs in CD8+ T Cell Subsets

DEGs were analyzed for different CD8+ T cell subsets. The screening criteria for DEGs were |log2FC| > 1, and the adjusted *p* value < 0.05. It was defined as DEGs set 1.

### 2.9. Functional Enrichment and Metabolic Pathway Activity Analysis of CD8+ T Cell Subsets

GO and KEGG enrichment analysis of DEGs of different CD8+ T cell subsets was performed using the R ‘clusterProfiler’ package. The screening criterion of GO terms was *p* value < 0.05. The criteria for KEGG pathway screening were minGSSize = 5, maxGSSize = 500, and q value Cutoff = 0.05. All canonical pathways (c2.cp.kegg.v7.4.entrez.gmt) were downloaded from the GSEA database, and metabolic pathway activities were calculated using the R ‘GSVA’ package. Hallmark gene sets (h.all.v7.4.entrez.gmt), CTLA4 signaling pathway gene sets (c2.cp.biocarta.v7.4.entrez.gmt), and PD1 signaling pathway gene sets (c2.cp.reactome.v7.4.entrez.gmt) were downloaded and the pathway activity was analyzed using R ‘GSEABase’ package.

### 2.10. Analysis of DEGs between High and Low Proportion of Exhausted CD8+ T Cells

The TCGA samples were divided into the high- and low-proportion groups according to the median of the exhausted CD8+ T cells ratio, and the R ‘limma’ package was used to calculate the DEGs between the two groups. The screening criteria for DEGs were |log2FC| > 1 and *p* value < 0.05. It was defined as DEGs set 2.

### 2.11. Construction of PPI Network and Identification of Hub Genes

The intersection of DEGs set 1 and DEGs set 2 was obtained. The PPI network was constructed through the string database (https://www.string-db.org/, accessed on 4 July 2022), and genes with a connection number greater than or equal to 5 were selected as candidate hub genes. Survival analysis was performed on candidate hub genes using the R ‘survival’ package, and the genes with a significant effect on prognosis were selected as hub genes.

### 2.12. GSEA Analysis of Hub Gene

The samples were divided into high- and low-expression groups according to the median hub gene expression value. KEGG pathway enrichment analysis was performed on the hub gene high- and low-expression groups using GSEA, and the top 3 pathways with significant differences were selected for mapping.

### 2.13. Clinical Sample Validation of Hub Genes

The surgical resection specimens and clinical information of 24 patients with LUAD who were newly diagnosed and surgically treated in Qilu Hospital at Shandong University (Qingdao, China) were selected. The expression levels of the hub genes in cancer tissues and adjacent tissues were analyzed through RT-qPCR, and four pairs of samples were randomly selected for Western blot detection. The protocol for using clinical samples in this study was approved by the Ethics Committee of Qilu Hospital of Shandong University (Qingdao, China).

## 3. Results

### 3.1. Single-Cell Clustering Results of LUAD Tissues

A total of 15 LUAD samples were obtained, 11 of which were obtained through surgical resection, named LUNG, and were part of AJCC stages I to III; 3 of which were obtained through endobronchial ultrasound, named EBUS, and all these were of AJCC stage IV; the last one was obtained through bronchoscopy, named BRONCHO, and it was from stage IV. The three groups of samples covered LUAD samples from the early to advanced stages. The number of genes, mRNA numbers, and the percentage of mitochondrial genes in the three groups are shown in Appendix A. The correlation coefficient between the proportion of mitochondrial genes and the number of mRNAs was 0.03 (Appendix A), and the correlation coefficient between the number of mRNAs and the number of genes was 0.89 (Appendix A), indicating that there is a significant correlation between the sequencing depth and the number of genes detected. Twenty principal components were obtained using the permutation test method based on the zero distribution, all of which were highly significant and could be used for subsequent analysis (Appendix A). A total of 22 clusters were obtained through TNSE clustering (Appendix A), and CD8+ T cells were mainly distributed in cluster 0, cluster 1, and cluster 2 (Appendix A).

### 3.2. CD8+ T Cells Clustering Results

After screening and filtering, pure CD8+T cells were again analyzed through TSNE clustering. A total of ten CD8+ T cell subsets were obtained, including eight cytotoxic CD8+ T lymphocyte (CTL) subsets, one naive-like CD8+ T lymphocyte (NTL) subset, and one exhausted CD8+ T lymphocyte (ETL) subset (Figure 1A).

### 3.3. Analysis of Differentiation Trajectory of Different CD8+ T Cell Subsets

As shown in Figure 1C, the expression profile of the NTL subset is typical, including naive-like markers CCR7, TCF7, LEF1, and SELL, and their expression levels gradually decreased with pseudo time as expected (Appendix A).

CTL subset 1 accounted for the largest proportion (Figure 1B), with a high expression of NKG7, GZMA, and CST7, and a moderate expression of CTSW, GZMB, and GZMH (Figure 1C). NKG7 is essential for the cytotoxic degranulation of natural killer (NK) cells and CD8+ T cells [10]. GZMA is secreted by effector cytotoxic T cells and NK cells and is significantly correlated with intratumoral immune cytolytic activity [11]. CST7 is a cysteine peptidase inhibitor known to be expressed in NK cells and CD8+ T cells during steady-state conditions [12]. The expression of CTSW is positively correlated with the infiltration level of immune cells, including CD8+ T cells in tumors [13]. The low level of GZMB is associated with poor prognosis in NSCLC patients treated with immune checkpoint inhibitor (ICI) therapy (PD-1 blocking) [14]. GZMH plays an essential role in NK-cell and T-cell-mediated cytolysis [15]. These results suggest that CTL subset 1 is in a cytotoxic activated state. In the cell differentiation trajectory, most of CTL subset 1 is in a different branch from the NTL subset and ETL subset. There are two separate branches, suggesting that CTL subset 1 may have a deeper heterogeneity (Figure 1D,E).

The marker gene expression profile of CTL subset 2 was similar to that of CTL subset 1. Still, it highly expresses GZMK (Figure 1C), a recently revealed marker for a specific subset of T cells with specific epigenetic and transcriptional characteristics. It develops in response to an aging host environment and expresses exhaustion markers [16]. This also causes CTL subset 2 to become confusing on the differentiation trajectory throughout the process of naive-like CD8+ T cell maturation to cytotoxic CD8+ T cells and become widely distributed in the pathway to exhausted CD8+ T cells (Figure 1D,E). CTL subset 3, as the third subgroup (Figure 1B), expresses GZMA and NKG7 at a moderate level and has the characteristics of cytotoxic CD8+ T cells. At the same time, it also expresses the naive-like marker IL7R, whose expression level is second only to the NTL subset (Figure 1C), which leads to its inability to concentrate on a specific branch in the differentiation trajectory (Figure 1D,E). CTL subset 4 and CTL subset 6 are unique, with high expression of cytotoxic markers GZMA and NKG7. At the same time, they have similar expression profiles to the ETL subset, such as exhaustion markers LAG3 and TIGIT (Figure 1C), which are also abundantly distributed at the end of the ETL subset in the cell differentiation trajectory (Figure 1D,E). It is speculated that they may be at a higher level of depletion. The expression profiles of CTL subset 5 and CTL subset 7 are similar, with high expression of NKG7 and GZMA, and moderate expression of PRF1 (Figure 1C), which seems to represent high cytotoxicity, but they have a significant overlap with the NTL subset and ETL subset in the cell differentiation trajectory (Figure 1D,E). Because of their small proportion, especially in CTL subset 7, their exact biological significance needs to be further studied. The proportion of CTL subset 8 was the smallest, and its expression profile and cell differentiation trajectory were most similar to those of CTL subset 2 (Figure 1C–E). The ETL subset expressed typical exhaustion markers LAG3, TIGIT, PDCD1, HAVCR2 and CTLA4 (Figure 1C). Their expression gradually increased with pseudo time (Appendix A–L).

Interestingly, the ETL subset also expressed GZMA and NKG7, suggesting that the exhausted T cells, as defined herein, are not entirely dysfunctional, as can also be seen from the pseudo time distribution of the displayed cytotoxicity markers PRF1, GZMA, GZMK, and NKG7, which were expressed throughout the cell differentiation process (Appendix A). LAG3 is a surface molecule found on immune cells, and recent studies suggest that LAG3 is a promising immune checkpoint that negatively regulates T cell activation [17]. TIGIT presents an earlier expression dynamic than PD-1 in activated CD8+ T cells and is upregulated in non-small-cell lung cancer patients. When anti-TIGIT mAb (tiragolumab) is used in combination with anti-PD-L1 mAb (atezolizumab), it shows better clinical effects than single drugs [18].

Naive-like CD8+ T cells are activated after antigen recognition and differentiate into SLECs or MPECs. The phenotypic heterogeneity of effector CD8+ T cells has been widely confirmed [19,20]. As shown in Appendix A, among the eight cytotoxic CD8+ T cells we defined, CTL6 and 7 are closer to SLECs (CD127^low^KLRG1^+^T-bet^high^CX3CR1^high^). Interestingly, CTL6 and 7, especially CTL7, rarely stay in the effect state but move directly from the naive-like state to the functional exhaustion state in cell differentiation trajectory analysis, which seems to be consistent with the short-term effect of SLECs. CTL3 appears to have the potential to develop into MPECs (CD127^high^KLRG1^-^T-bet^low^CX3CR1^low^), consistent with more CTL3 being distributed at the effector end. By observing the expression of TCF1, TOX, and PD-1 in the ten CD8+ T cell subsets, the ETL subsets we defined are closer to terminally exhausted CD8+ T cells (TCF1^-^TOX^+^PD1^high^). In comparison, nearly half of the eight cytotoxic CD8+ T cell subsets have the characteristics of progenitor exhausted CD8+ T cells (TCF1^+^TOX^-^PD1^low^).

### 3.4. Interaction Network and TFs Expression Analysis of Subsets

There were significant interaction pairs among the ten subsets (Appendix A), among which CTL subsets 4 and 7 and the ETL subset had relatively more interaction pairs (Appendix A), and CD74-MIF and KLRB1-CLEC2D were the two most frequent pairs (Appendix A). As a high-affinity membrane receptor, CD74 is involved in multiple signaling pathways mediated by macrophage migration inhibitor (MIF), including promoting the Warburg effect by activating the NF-κB/HIF-1α pathway in lung cancer [21]. A recent study also confirmed that MIF could be used as one of the predictors of lymph node metastasis and prognosis in LUAD [22]. CD161 encoded by KLRB1 was recently identified as a novel immune checkpoint molecule, and blocking the CD161-CLEC2D pathway strongly enhanced T cell killing against tumor cells and reduced T cell exhaustion [23]. Expression analysis showed that the ETL subset has more highly expressed TFs, mostly related to the inhibition of effector T cells, including NEAT1, ENO1, RELB, and so on (Appendix A). Among them, the down-regulation of NEAT1 was confirmed to inhibit CD8+ T cell apoptosis and enhance cytolytic activity through the miR-155/Tim-3 pathway [24]. Anti-ENO1 antibody inhibits myeloid-derived suppressor cells (MDSCs) infiltration in the TME and attenuates its inhibitory effect on effector T cells [25]. Notably, CTL subset 7 specifically highly expresses IRF1 and AES, of which IRF1 has been found to be associated with tumor suppressor activity in multiple studies [26]. CTL subset 3 specifically expresses ZFP36, an RNA-binding protein that can significantly inhibit T cell activation [27].

### 3.5. Functional Analysis of CD8+ T Cell Subsets

GO analysis was performed on CD8+ T cell subsets, as shown in Appendix A. CTL subsets 1, 4, and 7 overlap with naive CD8+ T cells, involving mRNA catabolism, signal recognition particle (SRP)-dependent co-translational protein and membrane transport, endoplasmic reticulum protein localization, viral gene transcription, and protein expression. CTL subsets 3, 5, and 6 are functionally similar, involving B cell activation, antigen receptor-mediated signaling, and lymphocyte differentiation. CTL subset 2 and 8 function relatively independently. CTL subset 2 function involves differentiation and activation of T and B cells, chemotaxis of monocytes, and stronger cell killing. CTL subset 8 focuses on immune response activation, which can respond to glucocorticoids and cAMP. The function of the ETL subset is unique, focusing on the relevant pathways of sugar and energy metabolism, including the metabolism and regeneration of NADH, the typical glycolytic pathway, and is involved in the differentiation and activation of lymphocytes.

The analysis results of KEGG and metabolic pathways are shown in Appendix A. These 10 CD8+ T cell subsets mainly focus on the pathways of pathogen infection, transplant rejection, antigen processing and presentation, Th1 and Th2 cells differentiation, and apoptosis.

In the analysis of hallmark and immune checkpoint pathway activity, the ETL subset has a high activity of allograft rejection, the IL-2/STAT5 signaling pathway, and the PD-1 signaling pathway. By contrast, the P53 signaling pathway was significantly inhibited (Appendix A).

### 3.6. Distribution of Different CD8+ T Cell Subsets in TCGA-LUAD Samples

The proportion of 10 CD8+ T cell subsets in immune cells was identified from 524 TCGA-LUAD tumor samples. The results are shown in Figure 2A. The ETL subset accounted for the largest proportion, followed by CTL subsets 3, 4, and 2. The proportion of CTL subsets 5, 7, and 8 was 0 or close to 0. The specific distribution heatmap of different subsets in the TCGA-LUAD samples is shown in Figure 2B.

The TCGA-LUAD samples were divided into high- and low-proportion groups according to the median proportion of each subset, and then, survival analysis was performed. The results showed that the proportion of the ETL subset had a significant impact on the overall survival (OS) of LUAD patients (*p* = 0.0098, Figure 2C). The higher the proportion, the worse the patient prognosis. The proportion of other subsets had no significant effect on the OS (Appendix A).

In addition, the proportion of the ETL subset showed a positive correlation trend with the T stage (Figure 2D). The higher the T stage, the greater the proportion of the ETL subset, but there was no significant correlation between the N stages, M stages, and AJCC stages (Appendix A).

### 3.7. Identification of Hub Genes

Compared with other subsets, there were 26 DEGs in the ETL subset (Figure 3A, Appendix A), which was the previously defined DEGs set 1. The TCGA-LUAD samples were divided into high- and low-proportion groups according to the ETL subset ratio. The differential genes were calculated to obtain 41 DEGs (Figure 3B, Appendix A), which was the previously mentioned defined DEGs set 2. A total of 65 DEGs were obtained by combining the two. Furthermore, the PPI network was obtained through a string database, and 19 hub genes were obtained by selecting the genes with more than or equal to five connections (Figure 3C, Appendix A). Survival analysis of 19 hub genes using the R ‘survival’ package showed that advanced glycosylation end-product specific receptor (AGER), CD69, glyceraldehyde 3-phosphate dehydrogenase (GAPDH), and IL7R had a significant effect on the OS of LUAD patients (Figure 4A–D). We validated the same differential expression of these four genes in an independent external dataset GSE43458, and the results were not related to smoking status in patients with LUAD (Appendix A). We also confirmed the same expression trend of the above four genes in our clinical tissue samples. RT-qPCR and Western blot results showed that the expressions of AGER, CD69, and IL7R in cancer tissues were significantly lower than those in adjacent tissues. At the same time, GAPDH was significantly higher in cancer tissues (Figure 3D,E). The clinical information of the 24 patients is shown in Appendix A. The primers and antibody-related data are shown in Appendix A.

### 3.8. GSEA Analysis of Hub Genes

AGER belongs to the immunoglobulin superfamily and shows low expression in the LUAD samples in this study. GSEA analysis results show that AGER is associated with arachidonic acid (AA) metabolism (Figure 4E and Appendix A). AA has a substantial regulatory effect on the immune system. AGER-mediated lipid peroxidation has been shown to drive caspase-11 inflammasome activation in sepsis [28], while inflammatory responses in tumor tissue are often favorable for patient prognosis. AGER has also been directly shown to have inhibitory effects on the development of lung cancer and is a potentially favorable prognostic marker for NSCLC [29].

CD69 can act as a CD8+ tumor-infiltrating lymphocyte (TIL) activation marker. Consistent with previous studies, CD69 expression levels were significantly reduced in LUAD samples in this study [30]. GSEA analysis showed that high expression of CD69 was associated with the chemokine signaling pathway, the T cell receptor signaling pathway, and natural killer cell-mediated cytotoxicity (Figure 4F and Appendix A).

GAPDH has received more attention due to its biological role in tumors. Knockout of GAPDH in human cancer cell lines results in cell proliferation arrest and resistance to S-phase-specific cytotoxic drugs [31]. This is consistent with our GSEA analysis of GAPDH, whose expression is closely related to the cell cycle (Figure 4G and Appendix A).

IL7R is associated with the risk of NSCLC, and IL7R deficiency induces an increase in tumor-infiltrating regulatory T cells [32]. In addition, IL7R is also significantly correlated with PD-1 expression, which may be a predictive marker for the PD-1 inhibitor response in patients with PD-L1-negative lung squamous cell carcinoma [33]. GSEA analysis showed that the interaction between cytokines and cytokine receptors, phagocytosis mediated by FcγR, and the JAK/STAT signaling pathway were the regulatory targets of IL7R (Figure 4H and Appendix A).

## 4. Discussion

The role of CD8+ T cells in tumor control has been well established. Compared with circulating and lymph-node-resident CD8+ T cells, there is still a lack of consistent conclusions about the source of CD8+ T cells that have infiltrated and been resident for a long time in the tumor. First, including the large numbers of naive-like CD8+ T cells infiltrating the LUAD demonstrated in this study, where do they come from? Unlike effector CD8+ T cells, which have a strong ability to reside in surrounding tissues, they usually circulate in the blood and lymphoid organs. Some researchers have pointed out that lymphoid aggregates in the tumor may be the source of these stem cell-like CD8+ T cells [34]. Second, do the infiltrating effector CD8+ T cells and exhausted CD8+ T cells invade from the circulatory system or evolve gradually from naive-like CD8+ T cells in the tumor? The current research conclusion tends to the latter, and the T cell receptor pedigree analysis also supports this conclusion [35].

In this study, by analyzing the differentiation trajectories of 10 CD8+ T cell subsets, we can see that the cells at the effector CD8+ T cell state accounted for only about 18% of the total number, the CD8+ T cells in the naive-like and exhausted state accounted for about 16% and 17%, respectively, and the remaining 49% or so of the cells were located between the lines connecting the three endpoints. On the one hand, this can explain the fact that the number of CD8+ T cells infiltrating tumors does not directly reflect its antitumor ability, and the proportion of CD8+ T cells with typical cytotoxic effects infiltrating LUAD is very low. On the other hand, it is confirmed that CD8+ T cells do not exist in a ternary state in LUAD but in a continuous transition process.

Interestingly, we found that some subsets started from naive-like CD8+ T cells and seemed to differentiate not towards effector CD8+ T cells but rather directly to the state of exhausted CD8+ T cells, such as CTL subsets 2, 5, and 7. What causes them to deviate from the typical differentiation trajectory remains to be further studied. At present, it is believed that antigen recognition in TME is an important driver of T cell dysfunction, and the continuous stimulation of complex antigen populations promotes the rapid development of T cell dysfunction phenotypes, including the decrease in cytotoxic markers and increased expression of inhibitory receptors.

By combining the cell differentiation trajectory and the expression of cell surface markers, we can roughly describe the dynamic evolution of CD8+ T cells, starting from naive-like CD8+ T cells, with CTL subsets 2 and 3 as the transition stage, passing through the intermediate activation state of CTL subset 8, reaching the effector T cell state of CTL subset 1, and then passing through the intermediate exhausted state of CTL subset 4 and 6, and finally reaching the exhausted state. The other part, such as CTL subsets 5 and 7, seldom go to the effector T cells through the above pathway; they rather go more directly into the exhausted state. Notably, it was difficult to find a well-defined subset in our clustering, such as CTL subset 1 at the apex of effector T cells, which still partially expresses GZMK, despite being considered a pre-exhausted marker. Other subsets of cells also have different “mixed” expression levels, suggesting the complexity and heterogeneity of CD8+ T cells in LUAD.

In this study, the proportion of the ETL subset significantly impacts the prognosis. The higher the proportion, the worse the prognosis, reflecting that T cell exhaustion affects the normal cytotoxicity of CD8+ T cells and then affects the prognosis of patients. The exhausted subset we define here is not a completely dysfunctional cell. It may be that the function has changed or even that new functions have emerged. For example, CD8+ T cells with early dysfunction gain higher proliferation ability, and CD8+ T cells with late dysfunction acquire the ability to produce CXCL13 [36], which induces the production of B cells. It is worth mentioning that some studies have shown that a small number of dysfunctional CD8+ T cells are the driving force for the durable response to immune checkpoint inhibitors [37]. One possible explanation is that these cells highly express PD1 and CTLA4, which are the direct anti-PD1 and anti-CTLA4 therapy targets. After treatment, they can restore the production of cytokines (TNF, IL-2, and IFN-γ) and cytotoxicity. However, this phenomenon seems limited to mild exhausted CD8+ T cells; it cannot be recovered once severe functional exhaustion occurs [38].

Predicting receptor–ligand pairing is essential, and successful interactions of key proteins mediate intercellular communication. In this study, the interaction network between subsets of cells was explored, and there were many interactions between CTL 4, 7, and ETL subsets. TF is an important regulator of gene expression. The ETL subset was significantly associated with numerous TFs, which could help elucidate the mechanism of functional exhaustion of the ETL subset.

At the end of the study, we screened and verified the expression of AGER, CD69, GAPDH, and IL7R, which provide new prognostic markers for LUAD patients.

## 5. Conclusions

In this study, we delineated the differentiation trajectories of CD8+ T cells infiltrating the TME of LUAD, revealing the heterogeneity and diversity of CD8+ T cells. Elucidating the dynamic evolution and functional exhaustion of CD8+ T cells will help to understand the different responses of patients to tumor immunotherapy and provide potential molecular targets for improving the effect of immunotherapy.

## Figures and Tables

**Figure 1 cancers-14-05183-f001:**
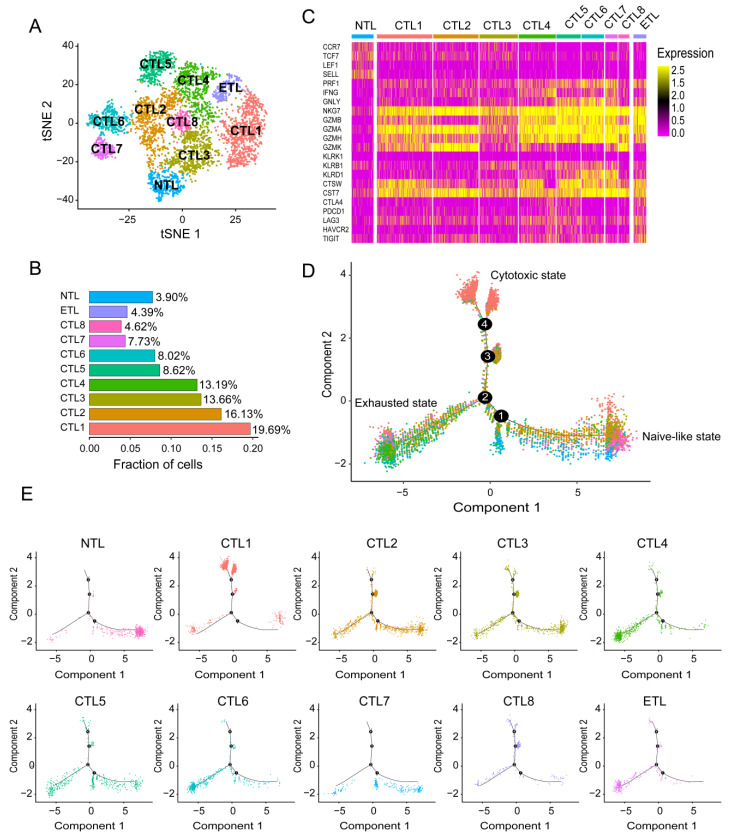
CD8+ T cell subsets and cell differentiation trajectories. (**A**) Cluster map of ten. CD8+ T cell subsets, including one NTL subset, eight CTL subsets, and one ETL subset. (**B**) Proportion of ten CD8+ T cell subsets. (**C**) Heatmap of marker gene expression for ten CD8+ T cell subsets. (**D**) Comprehensive cellular differentiation trajectories of ten CD8+ T cell subsets. A typical cell differentiation trajectory is from the naive-like state to the cytotoxic state, and then to the exhausted state. (**E**) Individual cell differentiation Trajectories for ten CD8+ T cell subsets.

**Figure 2 cancers-14-05183-f002:**
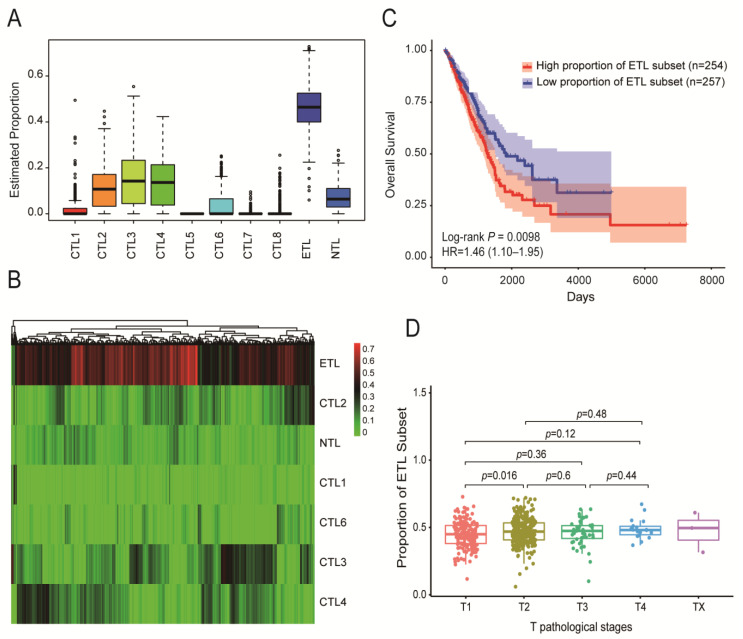
Distribution of ten CD8+ T cell subsets in TCGA-LUAD samples and their impact on patient prognosis. (**A**) Proportion of ten CD8+ T cell subsets among immune cells in TCGA-LUAD samples. (**B**) Heatmap of the distribution of ten CD8+ T cell subsets in TCGA-LUAD samples. (**C**) The effect of ETL subset proportion on the OS of LUAD patients: the higher the proportion of the ETL subset, the shorter the OS of LUAD patients. (**D**) Distribution of ETL subset in different T stages of LUAD patients: the larger the proportion of the ETL subset, the higher the T stage.

**Figure 3 cancers-14-05183-f003:**
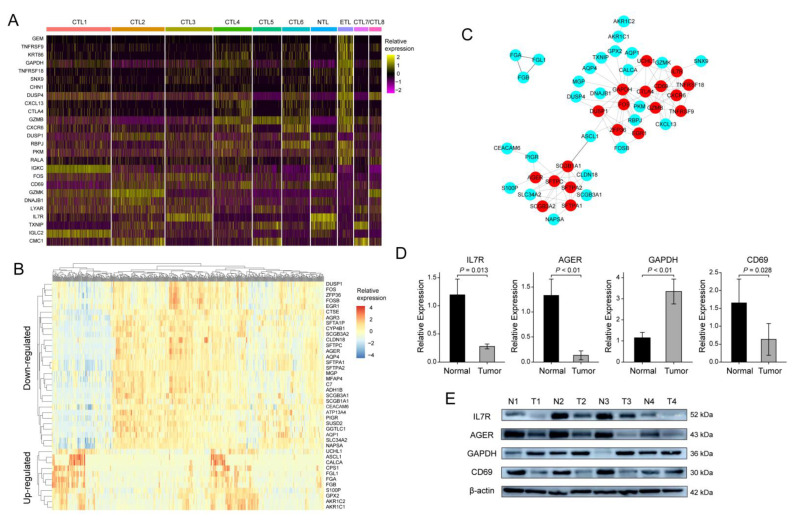
Screening of hub genes and validation of clinical samples: (**A**) 26 DEGs between the ETL subset and other CD8+ T cell subsets; (**B**) 41 DEGs among TCGA-LUAD samples with high and low proportions of ETL subset; (**C**) protein interaction network of 65 DEGs; (**D**) RT-qPCR results of IL7R, AGER, GAPDH, and CD69 in 24 pairs of LUAD samples; (**E**) Western blot results of IL7R, AGER, GAPDH, and CD69 in 4 pairs of LUAD samples.

**Figure 4 cancers-14-05183-f004:**
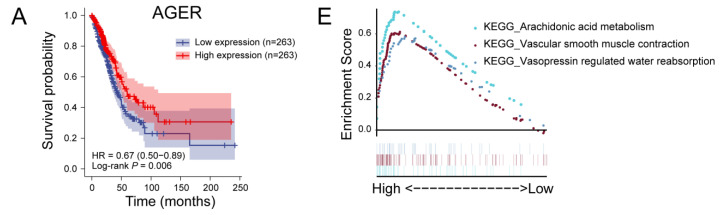
Prognostic analysis and single-gene GSEA analysis of hub genes. (**A**) Prognostic analysis of AGER. The higher the expression of AGER, the longer the OS of LUAD patients. (**B**) Single-gene GSEA analysis of the AGER. (**C**) Prognostic analysis of CD69. The higher the expression of CD69, the longer the OS of LUAD patients. (**D**) Single-gene GSEA analysis of the CD69. (**E**) Prognostic analysis of GAPDH. The higher the expression of GAPDH, the shorter the OS of LUAD patients. (**F**) Single-gene GSEA analysis of the GAPDH. (**G**) Prognostic analysis of IL7R. Before about 90 months, the higher the expression of IL7R, the longer the OS of LUAD patients. (**H**) Single-gene GSEA analysis of the IL7R.

## Data Availability

The GSE131907 dataset is available from the GEO database. TCGA-LUAD sample data can be obtained from the TCGA database. Additional information is available from articles or Appendix A. The codes used for analysis have uploaded to the GitHub website. Please access it from the following link: https://github.com/qilugaohaidong/R_script, accessed on 10 October 2022.

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
