# Peer review of "Heterogeneity and Differentiation Trajectories of Infiltrating CD8+ T Cells in Lung Adenocarcinoma"

_cancers, 2022, doi:10.3390/cancers14215183_

Round 1

Reviewer 1 Report

The authors have presented a very interesting study. but the study is not comprehensive. The authors have done a number of analysis but without biological validation, it is difficult to make any conclusions on this data.

Author Response

Thank you for your review of our manuscript. This study is indeed based on Kim et al.'s fruitful exploration of lung adenocarcinoma (PMID: 32385277). The authors selflessly uploaded extensive single-cell sequencing data of 58 samples from 44 patients to a public database (GEO: GSE131907). We have made secondary development and utilization of the data from different perspectives. Our focus in this study was to further identify CD8+ T cell subsets by rearranging and filtering out pure CD8+T cells using TSNE clustering; we finally obtained ten continuously changing CD8+T cell subsets. As described in the discussion, we observed that the cells at the effector CD8+ T cell state account for only about 18% of the total number, the CD8+ T cells in the naive-like and exhausted state accounted for about 16%, and 17%, respectively, and the remaining 49% or so of the cells were located between the lines connecting the three endpoints.

Strict quality control and a rigorous analytical process increased the reliability of the results. Our results at least partly explain the different response rates of LUAD patients to tumor immunotherapy. At the same time, the specific data support that it is challenging to rely on traditional immunohistochemistry to measure the number of tumor-infiltrating lymphocytes (without focusing on cell function) to predict whether patients respond to immunotherapy.

We screened for characteristic molecules of functional exhaustion (by analyzing the differential expression among subgroups, especially those related to prognosis, such as the four genes screened in this study). We also verified the expression of the above four genes in tissue samples by qPCR and WB. Following the reviewers' recommendations, we will continue to conduct experimental studies to explore the role of these key characteristic molecules in the process of CD8+T cell function depletion and the transformation of "cold" and "hot" tumors.

Reviewer 2 Report

The manuscript “Heterogeneity and differentiation trajectories of infiltrating CD8+ T cells in lung adenocarcinoma” by Song et al., describes the heterogeneity and functional exhaustion of infiltrating CD8+ T cells in LUAD. The author identifies set of genes that can be used as a prognostic marker in case of LUAD. The identification of differentiation trajectories of CD8+ T cells in tumor micro environment is interesting to read, and hence this paper has the potential to be important. The experiments were well designed, however there are still points to be confirmed.

For identification of cytotoxic CD8+ T cell clusters, it could be further sub-clustered into SLECs (Short-lived effector cells) and MPECs (Memory precursor effector cells). This clustering will further provide a more clear understanding on differentiation trajectories of infiltrating CD8+ T cells in LUAD. You may consider the markers such as T-bet, KLRG1, CX3CR1 etc.

For identification of exhausted CD8+ T cell clusters, it could be further subdivided into progenitor exhausted and terminally exhausted CD8+ subpopulations.

Reviewer 3 Report

The study by Song X et al. is well designed, conducted and described. The topic is innovative and cutting-edge. Methods are adequate and statistics appropriate. The characterization of lymphocytes dynamics (differentiation trajectories of CD8+ in TME) is crucial to ameliorate both the comprehension of lung adenocarcinoma (LUAD) biology and patients' treatment with immuno-therapy. The authors present genetic, phenotypic and prognostic analyses of their clinical dataset as well as of public datasets. The conclusions are consistent with the presented data and the study contributes to deep the insight into the heterogeneity and functional exhaustion of CD8+ T cells in TME of LUAD. The discussion is intriguing and prompts further research.

Author Response

Thank you for your review and positive comments on our manuscript.

Reviewer 4 Report

The article is of great interest. It is well written clearly and comprehensively

Section 2.4 needs clarification how the 10 subsets were identified within TCGA separately as CIBERSORT define CD8+ as one entity. Was a risk score used to combine the expression matrix of DEGs per cluster? It seems need more clarification in the text.

Fig S4C: It seems ribosomal genes were not filtered before starting the scRNA-seq analysis.

Recommend adding the github link for the codes used for analysis.

Fig.2A need y-axis title. Wonder why specifically ETL and testing the other subsets was used in Fig.2D. There was no significance in others?

The article provided good information. Suggest validating the 4 final markers in other GEO studies for LUAD.

TF and ligand receptor pairs could be mentioned in the discussion.

Round 2

Reviewer 1 Report

author's reply is satisfacory.